# Participation in an Online Course about Death and Dying: Exploring Enrolment Motivations and Learning Goals of Health Care Workers

**Claris Oh [1],*, Lauren Miller-Lewis [2,3]** and **Jennifer Tieman [1]**

[1] College of Medicine and Public Health, Flinders University, Adelaide, SA 5000, Australia; jennifer.tieman@flinders.edu.au

[2] Research Centre for Palliative Care, Death and Dying, Palliative and Supportive Services, College of Nursing and Health Sciences, Flinders University, Adelaide, SA 5000, Australia

[3] Department of Psychology and Public Health, School of Health, Medical, and Applied Sciences, CQUniversity Australia, Adelaide Campus, 44 Greenhill Road, Wayville, SA 5034, Australia; l.miller-lewis@cqu.edu.au

* Correspondence: claris.oh@flinders.edu.au

**Abstract:** The Dying2Learn massive open online course (MOOC) was a five-week course designed for the general community covering various topics related to death and dying, developed with a sociocultural focus that encouraged personal reflection and open discussion, rather than a medical or clinical focus. Yet, the majority of the participants identified as health care workers. Their motivations to enrolling in the course as well as their learning goals were examined. Responses to questions relating to their motivations for enrolment and learning goals were assigned codes and grouped into themes. We then made comparisons between the different demographic and occupational groups. The most commonly mentioned responses related to general interest in the topic of death and dying. HCWs were more likely to mention work-related motivations and improving communication skills than non-HCWs. We found that HCWs hoped to be better at talking about death and dying, which might indicate a possible gap in their formal education in this area.

**Keywords:** death and dying; health care worker; CPD; online learning; death attitudes; palliative care; end-of-life

## 1. Introduction

The topic of death is shunned by many and most are uncomfortable talking about it even though death comes to everyone [1,2]. Yet, it is essential to have a better understanding of death and dying to be able to prepare for it and learn to accept this natural phenomenon as something part and parcel of life [3]. This could be achieved through educational activities related to death, dying, and grief, known as death education [4]. Teaching such a sensitive topic in a traditional way can be tricky, stirring up unpleasant emotions or even causing students to withdraw from their emotions in order to avoid the discomfort of confronting death [5].

With this in mind, Dying2Learn was designed to allow the participants from the general community to cover various topics related to death and dying through a five-week online course known as a MOOC (massive open online course). Instead of just having educators deliver fixed course content to passive learners, participants of MOOCs are able to share and discuss content, through the online forums and comment on others' submissions and receive comments on their own. In a digital age, social learning and the ability to freely exchange information and ideas is increasing in popularity [6,7].

The Dying2Learn MOOC was developed by an Australian government-funded palliative care knowledge network known as CareSearch (www.caresearch.com.au) and was hosted on a platform developed by OpenLearning Global Pty Ltd. (Sydney, Australia) (www.openlearning.com), an Australian-based company. It first ran in 2016 and was subsequently repeated in 2017 and in 2018. The three main objectives that the MOOC creators hoped to achieve were as follows:

1. To provide a vehicle for the Australian public to openly discuss and learn about issues around living, death, and dying in a supportive environment.
2. To explore the opinions and views of Australians around death and dying.
3. To understand the effects of online learning and discussion through a MOOC on participants' attitudes towards death and dying [8,9].

The MOOC had four core modules and concluded with a module for reflection and course evaluation. Each of the modules were delivered by MOOC facilitators who also served a moderator role in the forum discussions. Module 1 had participants reflect on how society engages with the topic of death and dying, and the type of language that is used to describe it. Portrayal of death and dying in the media was discussed as part of Module 2. In Module 3, participants looked at the role of medicine in prolonging life and death, and death in the age of internet was the focus of Module 4. Participants would assess content like videos and articles that the facilitators uploaded on the platform and respond to topic reflection questions by posting comments on discussion boards.

Registration and participation rates for the 2016 MOOC showed that there was a willingness to explore and discuss issues around death and dying and feedback was positive [8].

Although marketing for the Dying2Learn MOOC was directed towards members of public, health care workers (HCW) formed a majority of the participants in the MOOC. Promotional materials clearly indicated that the focus of the course was not related to clinical practice and skills required for providing care for patients. Therefore, the health professional interest was somewhat surprising. Looking at the motivations behind their decision to participate in such a course could help understand what need was being addressed, whether they felt the need to improve on their competencies in handling death and dying issues, or if their interest reflected perceived shortcomings in skills and training.

This study aimed to investigate the motivations of participants enrolling in the course, especially that of HCWs and whether there were differences between participants from different professional groups or health care settings these HCWs worked within. Information about the needs of HCWs in respect to death and dying is particularly important given the projected increase in demand for palliative and end of life (EOL) care associated with ageing populations and changing patterns of disease burdens [10]. Multiple studies have already highlighted the lack of EOL care training for physicians and medical students specifically [11–13], with a developing body of literature looking at EOL care training for other HCW groups such as aged care workers or allied health staff [14,15]. Information about why they have elected to participate in an online course that addresses broad and personal considerations around death and dying could be particularly valuable in understanding HCW views, and how this relates to feeling comfortable or competent at handling such issues at work.

## 2. Materials and Methods

This project obtained ethical approval from Flinders University Social and Behavioural Research Committee (Project No. 7427), and consent from the participants was obtained prior to including their responses in this study.

We have chosen to use the data from the 2017 run of the Dying2Learn MOOC, as there were detailed HCW occupation data that we did not collect in the first run in 2016. The 2017 MOOC ran from the end of March to May with a total of 1960 enrolees, 783 of whom responded to activities in the introductory week of the MOOC. In this study, we examined responses to questions asking the participants why they have chosen to participate in the MOOC ($n = 772$ gave responses), and what their learning goals were ($n = 763$ gave responses).

Of the 783 enrolees who provided responses to one or both of these questions, 93.3% were female, and their age ranged from 18 to 82 (with a mean age of 48.75, *SD* = 12.18). For the purposes of analysis, we categorised the sample into three age groups: Age 18–39 (*n* = 178, 22.8%); age 40–59 (*n* = 448, 57.3%), and age 60–82 (*n* = 156, 19.9%). Forty percent of the 783 enrolees reported having previously done a MOOC, 58.2% had not, with the remaining unsure. These enrolees went on to complete an average of 67.12% of the MOOC course content (*M* = 67.12; *SD* = 34.43). Overall, 67% of the sample had completed university studies, and 82% resided in Australia. Of those living in Australia, 39.3% were living in rural/remote areas.

A total of 600 respondents reported having a health care worker (HCW) occupation, equating to 76.6% of the sample. Their stated occupational categories are listed in Table 1. Overall, *n* = 454 reported that they were currently working in an HCW role (58.1% of the total, and 75.67% of those who reported having ever having an HCW occupation).

**Table 1.** Occupation categories of respondents (*n* = 783).

|  | Frequency | Percent |
|---|---|---|
| Not a health professional | 183 | 23.4 |
| Aged care worker | 122 | 15.6 |
| Allied health | 112 | 14.3 |
| Doctor | 5 | 0.6 |
| Nurse | 282 | 36.0 |
| Other | 79 | 10.1 |
| Total | 783 | 100.0 |

In the first week of the 2017 course, participants were asked to complete a baseline survey, which helped to understand their socioeconomic background, including their occupations and whether they identified as being health care workers. Participants were also asked two open-ended questions, which produced text responses. Firstly, they were asked to "Briefly describe why you have decided to participate in the Dying2Learn MOOC." They were also asked to "Briefly describe what you hope to learn by completing the Dying2Learn MOOC." Responses to these two questions formed the basis of the current study to determine the motivations behind participation as well as the learning goals of the participants. After the MOOC concluded, the data that were collected were extracted for analysis.

In order to analyse the qualitative data, the text responses needed to be broken down into individual ideas that could be categorised, labelling these ideas with "codes". Codes are defined as a "researcher generated" word or short phrase that assigns meaning to the qualitative data. This coding process allowed for ideas to be mapped into major themes, and patterns could be identified as an indicator to how different groups of participants think [16].

The first author independently coded all responses of the two questions posed to the enrolees, assigning codes for individual ideas in each of the responses. These codes were descriptive to help summarise what the text response was saying. A second level of coding was then conducted through identifying repeated codes and identified themes.

To check for consistency in the coding data, responses for the two questions for 50 randomly selected participants were assigned to each of the two co-authors (L.M.-L., J.T.) for independent coding using a coding guide developed by the first author (C.O.). This determines whether all researchers identified the same themes from the responses, thus indicating high quality of the coding scheme and robust data evaluation [16]. We compared the assignment of codes for inter-rater reliability and calculated Cohen's Kappa to determine the proportion of inter-rater agreement over and above what would be expected from chance alone. Table 2 shows the results of the analysis and demonstrates good agreement rates between the independent raters.

**Table 2.** Inter-rater reliability agreement and Cohen's Kappa analysis.

| No. of Judgements Made | Total Disagreement | Agreement Rate | Cohen's Kappa |
|---|---|---|---|
| **C1 [1] vs. C2** | | | |
| Learning goals ($n$ = 490) | 35 | 92.86% | 0.675, $p < 0.0005$ |
| Why ($n$ = 490) | 39 | 92.04% | 0.652, $p < 0.0005$ |
| **C1 vs. C3** | | | |
| Learning goals ($n$ = 588) | 47 | 92.01% | 0.644, $p < 0.0005$ |
| Why ($n$ = 637) | 49 | 92.31% | 0.657, $p < 0.0005$ |

[1] C1 = Code 1; C2 = Coder 2; C3 = Coder 3.

After the final set of codes was applied to each participant's response to the 2 questions, we conducted quantitative descriptive statistical analyses to develop an understanding of the frequency of each theme in the data, and to determine whether the themes present in participants responses varied depending on their demographic characteristics and occupation.

## 3. Results

### 3.1. Responses Relating to Why Participants Enrolled in the MOOC

There were 772 responses for the question asking the participants to explain why they had decided to enrol in the MOOC.

Amongst the 14 themes identified, the most common motivation for participation was "Learn/ Understanding/Knowledge" ($n$ = 309, 39.5%). This was followed very closely by "Work/Education" ($n$ = 305, 39%) and "MOOC/Online/Accessible" ($n$ = 97, 12.4%). These are presented in Figure 1, with the more commonly mentioned themes highlighted. Definitions of the themes and the detailed results can be found in Table A1 in the Appendix A. We conducted Chi-square tests of Independence to determine the influence of various demographic factors on themes reported for why participants enrolled in the MOOC.

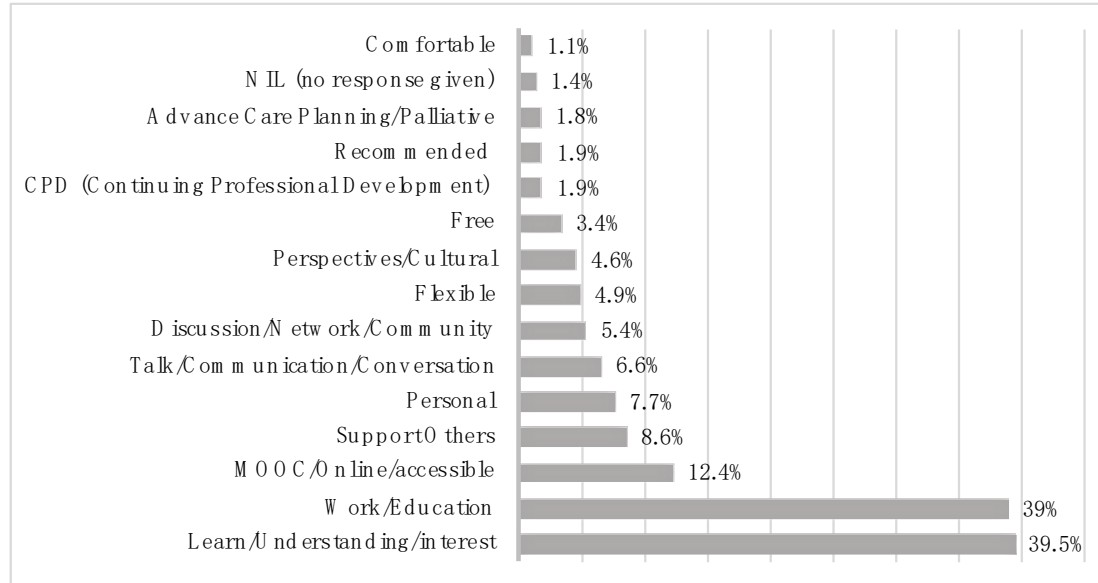

**Figure 1.** Percentages of participants who reported the different themes in response to why they participated in the massive open online course (MOOC).

We examined whether the themes present in their responses to why they enrolled in the MOOC varied based on the participant's gender and age group. However, the sample was predominantly female (93.3%), hence we were unable to make any conclusive remarks on any statistically significant differences between male/female responses. No age group differences were found to be statistically significant.

Next, we examined whether the themes present in their responses regarding why they enrolled varied based on whether the participant had an HCW occupation or not. HCWs were significantly more likely than non-HCWs to report work/education motivations (41.8% versus 29.5%; *Chi-Square* = 8.96 (*df* = 1), *p* = 0.003), and continuing professional development (CPD) motivations (2.5% versus 0%; *Chi-Square* = 4.66 (*df* = 1), *p* = 0.031), but were less likely to have discussion/networking motivations (3.5% versus 11.5%; *Chi-Square* = 17.57 (*df* = 1), *p* = 0.0001). CPD involves participating in learning activities that contribute to advancement of professional skills, and is required by the Australian Health Practitioner Regulatory Agency (AHPRA) for HCW to maintain their registration [17].

We examined the themes in motivations for why participants said they were enrolling for the three main health professional groups represented in this study. Participants were grouped into three categories based on occupation: Aged care worker (*n* = 122), allied health professional (*n* = 112), or nurse (*n* = 282). Other health workers (unspecified occupation), and those with an occupation outside of health were excluded from this analysis, as were doctors because the number of doctors (*n* = 5) was too small for the presence of adequate cell counts in the cross-tabulation analysis. These three health professional groups, along with non-HCWs, are presented in Table 3, along with the percentages for the themes they reported. Nurses were more likely than other groups to report motivations related to 'perspectives' (*Chi-Square* = 10.18, *df* = 2, *p* = 0.006). Aged care workers and Allied health workers more commonly reported work/education motivations (*Chi-Square* = 5.83, *df* = 2, *p* = 0.05) and supporting others as motivation (*Chi-Square* = 6.24, *df* = 2, *p* = 0.044). Allied Health professionals were more likely to report personal motivations than others (Chi-Square = 9.53, df = 2, *p* = 0.009).

**Table 3.** Themes reported by the different health professional groups and non-health care workers (HCWs) in response to why they participated in the MOOC.

| Themes | Not a Health Worker % | Aged Care Worker % | Allied Health Worker % | Nurse % | Total HCW Overall % |
|---|---|---|---|---|---|
| Learn/Understanding/Interest | 43.2 | 37.7 | 34.8 | 40.1 | 38.4 |
| Work/Education | 29.5 | 47.5 | 46.4 | 36.5 | 41.3 * |
| MOOC/Online/Accessible | 10.9 | 10.7 | 13.4 | 13.1 | 12.6 |
| Support Others | 8.2 | 13.1 | 4.5 | 7.4 | 8.1 * |
| Personal | 7.1 | 5.7 | 15.2 | 6.4 | 8.1 * |
| Talk/Communication/Conversation | 8.7 | 5.7 | 7.1 | 5.3 | 5.8 |
| Discussion/Network/Community | 11.5 | 2.5 | 2.7 | 4.3 | 3.5 |
| Flexible | 3.3 | 3.3 | 6.3 | 6.4 | 5.6 |
| Perspectives/Cultural | 6.0 | 0 | 1.8 | 6.0 | 3.7 * |
| Free | 3.8 | 1.6 | 4.5 | 3.5 | 3.3 |
| CPD (Continuing Professional Development) | 0 | 0.8 | 2.7 | 3.5 | 2.7 |
| Recommended | 1.6 | 0.8 | 2.7 | 2.8 | 2.3 |
| Advance Care Planning/Palliative | 1.1 | 2.5 | 0.9 | 2.5 | 2.1 |
| NIL (no response given) | 1.6 | 1.6 | 2.7 | 1.1 | 1.6 |
| Comfortable | 0 | 2.5 | 1.8 | 1.1 | 1.6 |

* Statistically significant difference between groups found.

For the *n* = 454 participants who reported currently working in an HCW role, *n* = 451 disclosed their work setting. Within this *n* = 451, 12.6% worked in a hospice or specialist palliative care, 21.3% worked in a hospital, 16.2% worked in primary or community care, 42.8% worked in residential aged care settings, and 7.1% reported working in other settings. We examined if their motivations for enrolling in the MOOC differed depending on the health setting that they reported working in, but no significant differences were found.

Of the sample, 70.1% of respondents reported having experience caring for a dying person in a health professional capacity, compared to 28.2% who did not, and 1.7% (*n* = 13) who were unsure. Work/education motivations were less frequently mentioned by those who reported having no experience caring as a health professional (30.3% compared to 42.4% for those who have and 38.5% for those unsure if they have, *Chi-Square* = 9.74 (*df* = 2), *p* = 0.008). Those with experience

in providing professional care for a dying person were more likely to mention CPD motivations (2.7% compared to 0.0% for the others, *Chi-Square* = 6.52 (*df* = 2), *p* = 0.038), and were less likely to mention discussion/networking motivations (3.3% compared to 10.0% for those who had not previously provided professional terminal care, and 15.4% for those unsure, *Chi-Square* = 16.45, (*df* = 2), *p* = 0.0005).

### 3.2. Responses Relating to Learning Goals

There was a total of *n* = 763 responses for the question requiring participants to share their learning goals for this MOOC. Thirteen themes were derived from the data collected and these are presented in Figure 2. Definitions of the themes and the detailed results can be found in Table A2 in Appendix A.

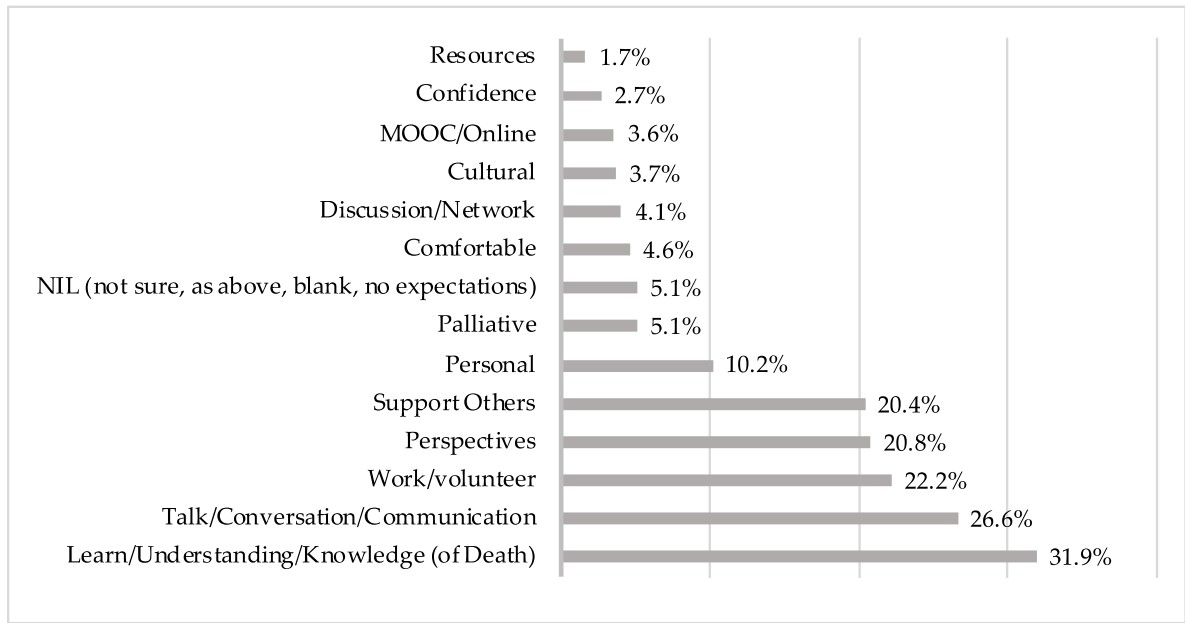

**Figure 2.** Percentages of participants who reported the different themes in response to learning goals.

The most commonly mentioned learning goal themes were "Learn/understanding/knowledge of death" (*n* = 250, 31.8%), followed by "Talk/conversation/communication" (*n* = 208, 26.6%) then "Work/volunteer" (*n* = 174, 22.2%). "Exploring new perspectives" (*n* = 163, 20.8%) and being able to "Support others" (*n* = 160, 20.4%) were also common learning goals. We conducted Chi-square tests of independence to determine the influence of various demographic factors like gender and age on themes reported for enrolees' learning goals. Again, we were unable to comment on differences between females and males because of the predominantly female sample (93.3%).

In comparing the responses between the different age groups, we found that participants aged 18–39 were significantly more likely to report palliative learning goals than the two older age groups (10.1% compared to 4.0% and 2.6% respectively, *Chi-Square* = 12.36 (*df* = 2), *p* = 0.002). Whilst not reaching statistical significance (likely due to low expected counts in two cells), there was also indication that the younger participants aged 18–39 were more likely to report gaining confidence as a learning goal (5.1% compared to 2.2% of those aged 40–59, and only 1.3% of those aged 60 or more, *Chi-Square* = 5.36 (*df* = 2), *p* = 0.069). No other age group differences were found.

Next, we examined whether there were variations in the learning goal themes present in their responses between participants with an HCW occupation or not. Non-HCW participants were significantly more likely to report a personal reason for their learning goal than HCWs (14.6% compared to 8.9%, *Chi-Square* = 5.06 (*df* = 1), *p* = 0.024). Coincidingly, HCWs were significantly more likely than non-HCWs to report work/volunteering learning goals (24.9% versus 13.5%; *Chi-Square* = 10.63 (*df* = 1), *p* = 0.001), talk/conversation/communication learning goals (28.4% versus 20.5%; *Chi-Square* = 4.51 (*df* = 1), *p* = 0.034), and palliative learning goals (6.4% versus 1.1%; *Chi-Square* = 8.11 (*df* = 1), *p* = 0.004).

We examined the themes in learning goals for the three main health professional groups, along with the non-HCW responses, as described in the previous section. These groups and comparisons are presented in Table 4.

**Table 4.** Themes reported by the different health professional groups and non-HCW in learning goals.

| Theme | Not a Health Worker % | Aged Care Worker % | Allied Health Worker % | Nurse % | Total HCW Overall % |
|---|---|---|---|---|---|
| Learn/Understanding/Knowledge (of Death) | 34.6 | 34.4 | 30.4 | 29.8 | 31.0 |
| Talk/Conversation/Communication | 20.5 | 27.0 | 27.7 | 29.1 | 28.3 |
| Work/volunteer | 13.5 | 23.0 | 21.4 | 27.0 | 24.8 |
| Perspectives | 23.8 | 12.3 | 24.1 | 21.3 | 19.8 * |
| Support Others | 18.4 | 22.1 | 18.8 | 21.6 | 21.1 |
| Personal | 14.6 | 10.7 | 12.5 | 7.1 | 9.1 |
| Palliative | 1.1 | 9.8 | 6.3 | 4.3 | 6.0 |
| Comfortable | 5.4 | 4.9 | 8.0 | 3.9 | 5.0 |
| Discussion/Network | 5.4 | 0 | 2.7 | 3.9 | 2.7 |
| Cultural | 2.2 | 2.5 | 5.4 | 4.6 | 4.3 |
| MOOC/Online | 5.4 | 1.6 | 6.3 | 2.5 | 3.1 |
| Confidence | 2.2 | 1.6 | 6.3 | 2.1 | 2.9 |
| Resources | 2.2 | 2.5 | 2.7 | 1.1 | 1.7 |
| NIL (not sure, as above, blank, no expectations) | 4.3 | 4.1 | 4.5 | 5.7 | 5.0 |

* Statistically significant difference between groups found.

It can be seen that Allied Health Professionals and Nurses were more likely than aged care workers to report learning goals related to perspectives (*Chi-Square* = 6.03, (*df* = 2), *p* = 0.049). No other differences between the three health professional groups reached statistical significance.

We examined if their learning goals differed depending on the health setting that they reported working in, but no significant differences were found.

## 4. Discussion

In this exploratory study, we aimed to determine the motivations behind people deciding to enrol in an online course about death and dying. The themes present in the responses to these two questions showed considerable overlap. Given the similarities in responses to the two questions, the results are discussed together below.

Participants most commonly mentioned general knowledge and learning about death and dying as both their motivation to join the MOOC as well as a learning goal. This suggests that there is genuine interest in engaging with issues around palliative care, death, and dying by both health care workers and non-health care workers, as well as ability to perform better at work. Some 40% of the participants reported having done a MOOC before, suggesting that they had pleasant learning experience through the platform. The Dying2Learn MOOC was first conducted in 2016 and it was also received positively with high satisfaction scores [9]. This result suggests that participants are open to learning on MOOCs and they are a feasible and useful learning vehicle. Indeed, there were participants who mentioned the nature of course delivery as a motivation to enrol in the course. MOOCs are not only free to access, but they also cater to busy schedules of participants as they allow them to complete the course flexibly in their own time.

Given that a little more than half of the participants identified as currently working in an HCW role, it was not surprising to find that work-related motivations and learning goals were amongst the most commonly mentioned. Both questions also prompted similar themes including wanting to improve communication skills and to explore new perspectives through open discussion on the online forum.

### 4.1. Differences between Demographic Groups

We wanted to determine whether there was a difference in motivations and learning goals between respondents of different demographic groups. However, as 93.3% of the sample identified as female, significant differences between the gender groups may be hidden. It could, however, be an indicator of females being more open to learning more about death and dying.

Amongst the different age groups, the younger 18–39 age group was more likely to report gaining confidence as a learning goal as compared with the older age groups. Older participants are more likely to have more exposure to issues of death and dying, as many would only experience death in their middle age [2]. Younger participants may lack experience and therefore seek training to feel more confident in their ability to talk about or handle issues regarding death and dying. This aligns with other studies that have shown positive correlation of age to perceived level of confidence. [18,19].

### 4.2. HCW Responses

The majority of the MOOC enrolees (76.6%) identified as having a health care worker occupation. HCWs were less likely to mention personal motivations than non-HCWs, suggesting that these participants were seeking to improve on their clinical practice rather than for the general objectives of personal reflections that the MOOC set out for them. Similarly, we found that HCWs were significantly more likely than non-HCWs to report work/education motivations and CPD motivations. Being health care professionals, they theoretically would have a relatively high level of death exposure as compared to the general public and are expected to be able to handle issues on death and dying proficiently. This could mean that HCWs seek additional training and instruction further to that provided in their professional education. It could also suggest that HCWs are seeking a different form of knowledge that builds confidence in managing complex and personal human issues as part of their health professional work. In addition, the higher level of exposure to issues of death and dying could make the HCW more aware of the importance of learning more about the topic and therefore have them seek out opportunities to do so.

Completion of the MOOC would also provide participants with a certificate of participation, which could provide HCWs with evidence to demonstrate engagement with training opportunities for professional development [20,21].

Being better at talking about death and dying was one of the most commonly mentioned learning goals amongst participants. This is an interesting finding as communication was not specifically promoted as one of the objectives of the MOOC. However, the MOOC does provide a platform for participants to talk openly about their reflections on the topic of death and dying. Having this safe environment to practice talking about it could, in turn, encourage participants to talk more about it outside of the course.

We also found that HCWs are significantly more likely than non-HCWs to state this as a learning goal. HCWs are more frequently exposed to death and dying in their jobs and thus would need to discuss issues surrounding the topic with their patients and the patients' next of kin. This is especially so if the HCW worked in EOL (end of life) care, palliative care, or even aged care services. Communication skills are considered to be one of the core competencies of EOL care. Better communication between patient and physician has also been shown to lead to better patient satisfaction [22]. On the flipside, poor communication is one of the most common (after care provision/treatment issues) complaints from health care consumers [23]. Unfortunately, research suggests HCWs do not perceive themselves as competent in conducting discussions about death and dying. Physicians report a lack of preparedness or even reluctance in conducting EOL conversations as they found the conversations uncomfortable, and difficult to manage [24]. Although they express a strong desire to learn more about the topic, few report receiving adequate training in communication about EOL issues [25,26]. A survey done with both nursing and medical students revealed that they did not know what to say in such situations, and had concerns that they would say the wrong thing [27]. This was concerning as it raises the

question on the adequacy of professional training in communication. Responses from our study suggest that our HCW participants have the same sentiments.

Interestingly, HCWs were less likely to have discussion/networking motivations as compared to those who had never been an HCW. In recent years, the importance of discussing matters surrounding death has been increasingly emphasised [28,29], yet death and dying has long been considered as a "taboo topic" that most in the public are uncomfortable with [1,2,30]. Non-HCWs may not have contacts that would readily participate in such conversations. On the other hand, given that HCWs have a higher level of exposure to issues of death, they could have had more opportunities to discuss the topic with their co-workers.

We were also interested in finding out whether participants from different health care professions had different motivations or learning goals. Due to the limited numbers in some of the professions (for example, doctors), we only compared the groups of "allied health worker", "nurses", and "aged care workers". Aged care workers and allied health significantly more commonly mentioned work/education motivations as compared to nurses. This could suggest that the nurses' professional training program might have provided adequate training when it comes to EOL care and communication, as compared to aged care workers and allied health workers. Another possible explanation is that nurses are more likely to be directly involved in providing care for EOL patients, and experience improves confidence and competencies in handling issues of death and dying [31]. Aged care workers more commonly mentioned supporting others as a learning goal than the other two groups of HCWs, which could indicate the difference in their roles as a carer.

We noted that there was no significant difference in motivations or learning goals for the HCWs from different health care settings.

### 4.3. Doctors' Responses in the MOOC

It was difficult to draw any conclusions from the responses of the participants who identified as doctors, due to the small numbers. We were interested in studying the doctors' responses as we thought it could be worthwhile to understand their motivations to enrol in the course and possibly identify gaps in EOL or communication training for doctors. Personal attitudes towards the topic of death and dying have been shown to affect patient care and communication between the physician and the patient [32]. HCWs usually refer to doctors as leaders of EOL care communication and thought of doctors as the ones responsible for initiating discussions with patients and next of kin. However, we should also recognize that non-physician HCWs also play important roles in facilitating EOL care communication [33]. A previous study on the 2016 Dying2Learn MOOC showed that following the MOOC, participants developed personal insight and better understanding to the topic of death and dying. More importantly, participants also became more comfortable discussing death [8]. It is hopeful that for doctors (and certainly other HCWs) participating in the course, this could translate into better patient care.

### 4.4. MOOCs as a Potential Learning Vehicle for Death Education

Some 40% of the participants reported having done a MOOC before, suggesting that they had pleasant learning experience through the platform. The Dying2Learn MOOC was first conducted in 2016 and it was also received positively with high satisfaction scores [9]. This result suggests that participants are open to learning on MOOCs and they are a feasible and useful learning vehicle. Indeed, there were participants who mentioned the nature of course delivery as a motivation to enrol in the course. MOOCs are not only free to access, but they also cater to busy schedules of participants as it allows them to complete the course flexibly in their own time.

This raises the question on whether MOOCs could be a potential form of training for HCWs to build up skills in dealing with issues of death and dying. MOOCs use collaborative learning model where students learn from each other through sharing experiences [34]. Given that HCWs are at the frontline of providing EOL care, the importance of death education for them cannot be understated.

Caring for the dying is perceived as a distressing experience [35], and could prompt HCWs to think about their own deaths as well. It is hence important to adequately prepare HCWs or even students in training to be emotionally ready as well to handle these issues.

However, the lack of control over what participants might gain from attending the course could limit the efficacy of MOOCs in providing EOL training for HCWs [36]. Furthermore, high satisfaction scores do not correlate with retention [37], which may suggest that MOOCs are not the best way for HCWs to receive essential competency training.

*4.5. Limitations*

There are some important limitations to this exploratory study that we should acknowledge. This study mainly utilised qualitative data, which may not produce results that are as robust as quantitative data. Although we employed exploratory quantitative analysis, it included multiple statistical tests which may have increased the likelihood of chance significant findings. In future studies, we could consider quantitative data collection to support these results derived from qualitative research methods.

The sample was also skewed towards a large majority of female participants, leading to an underrepresentation of males. The findings therefore may not be applicable to males in the general population. Amongst the HCWs, there was a limited number of doctors and other health workers and therefore we were not able to draw conclusions for these groups.

## 5. Conclusions

For an online course that was catered for the general public with a focus in encouraging open discussions about death amongst the participants, it was interesting to see that HCWs were motivated to enrol in the course. Studying their motivations and learning goals allowed us to understand their feelings towards death and dying, and also recognise where the potential gaps in HCW training for EOL care were. This knowledge could inform future development of similar courses or resources targeted at HCWs professional training. Talking about death and dying is not easy for anyone. While more research will need to be done to ascertain the efficacy of MOOCs in providing HCW skill competency training, a safe environment that a facilitated MOOC provides could help HCWs build up the confidence to talk about the topic in their work lives, improving their clinical practice.

**Author Contributions:** Conceptualization, C.O., L.M.-L., and J.T.; data curation, C.O., L.M.-L., and J.T.; formal analysis, L.M.-L.; investigation, C.O.; methodology, C.O., L.M.-L., and J.T.; supervision, L.M.-L. and J.T.; validation, L.M.-L. and J.T.; writing—original draft, C.O.; writing—review and editing, C.O., L.M.-L., and J.T. All authors have read and agreed to the published version of the manuscript.

**Funding:** This research study received no external funding. CareSearch is funded by the Australian Government Department of Health.

**Conflicts of Interest:** The authors declare no conflict of interest.

# Appendix A

**Table A1.** Themes derived from responses of enrolees (*n* = 772) to the question on why they participated in the Dying2Learn MOOC and examples of verbatim responses containing these themes.

| Theme | Definition | Examples | Number of Participants with This Theme Present in Their Response | Percentage of Participants with This Theme Present in Their Response (%) |
|---|---|---|---|---|
| Learn/Understanding/ Interest | Motivated by an interest on the topic, the course content, to learn more about the process of death and dying and gaining a better understanding | "Would like to learn more about death and dealing with persons dealing with death" <br> "Death and dying interests me and I want to learn more." <br> "I am fascinated by death. Been reading "Staring at the sun" by Irvin Yalom and it has re-ignited my interest" | 309 | 39.5 |
| Work/Education | Motivated by desire to perform better in their roles at the workplace—Includes wanting to improve on clinical practice, communication skills, supporting patients better, or seeking a career associated with death and dying | "Death and dying are of particular interest to me because I work in aged care." <br> "I wanted to up skill and also be better prepared in my job and have a better way to help our clients and their carers prepare for the final stage" <br> "I'm considering a career in Palliative Care and want to gain a better understanding of how death affects other people and hopefully what I can do to better support those facing it." <br> "I am currently establishing a new career as an End of Life Doula, having recently become a Funeral Celebrant. I saw the Dying2Learn course available through a Facebook page I follow." | 305 | 39 |
| MOOC/Online/accessible | Interest in participating and completing an educational course online, including the ability to participate in the course wherever they are located. Also includes responses with previous experience in online learning leading to further interest in participating in more of such courses | "It is convenient to do from home, I have never experienced this type of online or interactive learning, I usually avoid this! Feel I will benefit from learning in this way a new experience. Always keen to add to my knowledge of death and dying." <br> "I did the University of Tasmania's MOOC on Dementia three years ago and found it a wonderful way to learn about a field both of personal and professional interest." <br> "This course has come recommended to me by my facilities educator. I like that it is online and can be completed in my own time at my own pace. I also love that the group has many people with different jobs from nurses to carers giving various aspects on death and dying." | 97 | 12.4 |

**Table A1.** *Cont.*

| Theme | Definition | Examples | Number of Participants with This Theme Present in Their Response | Percentage of Participants with This Theme Present in Their Response (%) |
|---|---|---|---|---|
| Support Others | Seeking to pick up skills to support others who are going through or experiencing death, helping them to cope with death | "Have wanted to become more comfortable with the subject so I can be more supportive when friends and family are going through the end of life process themselves or with a loved one." "I'm interested as to why people are so frightened of the topic of death, what happens when we die, why people react differently to death, how can I best comfort someone during a palliative stage and the different outlooks on death." | 67 | 8.6 |
| Personal | Seeking an opportunity to reflect on their personal views on death and dying | "I want to participate to learn all I can about death and dying. I need to accept that death is going to happen and I want to conquer my fear of death and hopefully I can help others accept and not be afraid." "I have end stage lymphoma, and am now in palliative care. I am not afraid of death, and my excellent Pall Care team gives me a great deal of reassurance that I need not fear the dying process. However, I have difficulty discussing this with family and friends who say 'Oh, you'll live to be 100' Or 'Stay positive, don't talk about dying' and other such unhelpful things. I'd like to have some tips on how to better navigate these discussions." | 60 | 7.7 |
| Talk/Communication/ Conversation | To learn how to talk about death and dying with others, to be better at conducting these "difficult" conversations, to know what to say in these situations, and make better responses when confronted with death and dying | "To help to become more confident with open discussions about death and dying" "I want to be able to respond appropriately to everyone I ever converse with, about whatever we wish to discuss." "Since I work as a Palliative Care Psychologist for advanced cancer patients I come across deaths of our patients. Sometimes I feel some blocks on me while talking about death to few patients and caregivers. I am looking forward to this program to develop my skills in dealing with my problems with talking about death and help patients and caregivers." | 52 | 6.6 |

**Table A1.** *Cont.*

| Theme | Definition | Examples | Number of Participants with This Theme Present in Their Response | Percentage of Participants with This Theme Present in Their Response (%) |
|---|---|---|---|---|
| Discussion/Network/ Community | Motivated by the desire to talk about death and dying, and discuss the topic with like-minded people on the internet–hearing others' views on the matter as well as sharing their opinions | "I am interested in discussing death and dying and discovering and exploring different persons views on this topic. I am anticipating that this course will attract a diverse group of individuals that are willing to share their views, experiences and ideas on death and dying. I am sure this will assist in improving my knowledge and expertise in this area and support not only my personal development but also my professional." <br> "After being retrenched from my job of 12 years my path has taken an unexpected way into which is death and the dying. I want to know as much as I can about dying and every aspect and the only way to learn is to study and ask people. The MOOC has given me the opportunity to do both at the same time and to chat to like-minded people." | 42 | 5.4 |
| Flexible | Motivated by the ability to complete the course at their own pace, without needing to adhere to a fixed class schedule. | "It's an interesting topic where there is not much information about yet. Studying online on my own pace works for me, with my busy schedule and daily life." <br> "I felt that it might be beneficial in my practice as an Aged Care RN. The idea of being able to participate in an online format appealed because I thought it might be easier to fit into my schedule than other formats." | 38 | 4.9 |
| Perspectives/Cultural | Seeking to learn more about how people from different cultures view and deal with death and dying. To be able to expand their view on the topic, to see new perspectives as they hear from other participants in the MOOC | "Always interested to learn from others experiences and reflect on practice." <br> "I am interested in open discussion around dying and death as well as to learn as much as I can about this subject. When I saw this course through a Facebook link, I decided it was a great opportunity to engage with my own thoughts and feelings around dying and death as well as to hear the views of others. It is also a great opportunity to connect with a diverse community." | 36 | 4.6 |

**Table A2.** Themes derived from responses of enrolees (*n* = 763) to the question on their learning goals from participating in the Dying2Learn MOOC and examples of verbatim responses containing these themes.

| Theme | Definition | Examples | Number of Participants with This Theme Present in Their Response | Percentage of Participants with This Theme Present in Their Response (%) |
|---|---|---|---|---|
| Learn/Understanding/ Knowledge (of Death) | To learn more about the topic of death, understanding what death and dying means and the process of dying, to improve on their knowledge on the topic | "Having a better understanding of death and dying and seeing how other people react and feel." "Explore and clarify thoughts and expectations around death and dying for the person and their loved ones, before and after death." | 250 | 31.9 |
| Talk/Conversation/ Communication | To improve on communicating with others about death and dying, to make better conversations with others about death and dying | "Being more open to the discussions with families about the journey their loved one is on. Improve concepts of death and dying and improve my communication skills in these discussions." "I am hoping for more understanding of peoples' ideas about death and dying and to learn how to better communicate when needed with others." "I want to be able to talk openly about death and dying and not feel that I am talking about a taboo subject." | 208 | 26.6 |
| Work/volunteer | To gain skills to perform well or better in their roles at work, or volunteering activities | "How best to serve and support my Palliative Care clients." "I'm not entirely sure. I am hoping to have some really interesting conversations and hopefully come away with more knowledge about an industry that I would really like to work in." "Other strategies to speak with residents and families about end of life" | 174 | 22.2 |
| Perspectives | To be able to expand their view on the topic, to see new perspectives as they hear from other participants in the MOOC | "A better understanding of how different people feel about death. Maybe learn how best to approach the subject personally and at work. Broaden my understanding and appreciation of other perspectives." "I am inspired to be surrounded (even virtually!) by like-minded people. I am passionate about Palliative care and I aim to continue broadening my understanding and perspectives to better hold space and provide care to my clients and their families." | 163 | 20.8 |
| Support Others | To be able to apply what they have learnt in the course to support people who are going through or experiencing death, helping them to cope with death | "I hope that, with a better understanding of death and dying, I would be better able to journey with these people-to ally their fears, help them to accept their situations, bring some positives to their final days." "To improve my knowledge and understanding and to be able to support people at this time in life, to be able to initiate open discussion around death and dying and to reduce any fear, stress or anxiety." | 160 | 20.4 |

**Table A2.** *Cont.*

| Theme | Definition | Examples | Number of Participants with This Theme Present in Their Response | Percentage of Participants with This Theme Present in Their Response (%) |
|---|---|---|---|---|
| Personal | To have an opportunity to reflect on the topic in a personal capacity, and explore what their own views on death and dying | "What I hope to learn is how to become more accepting of the process as a person and to be more comfortable talking about death and dying when dealing with those on Palliative and their families."<br>"How to approach death in conversations and in reflection"<br>"I don't know. I am open to being challenged in the way I work but also for myself personally in how I approach it for myself." | 80 | 10.2 |
| Palliative | To learn more about palliative care specifically | "I hope to understand the palliative care aspect of EOL in more depth, & to have a greater understanding of the palliative care lexicon."<br>"I had done a Palliative Care course four years ago, I have since looked after a friend who had terminal cancer. It's always good to know new information in Palliative Care" | 40 | 5.1 |
| Comfortable | To feel comfortable with talking about dying, being around the idea of death and to engage people who are at the end of their lives-The main aim is to eliminate feelings of anxiety when conducting such conversations and thus be better conversationalists at a personal level. | "Death and discussing it makes people uncomfortable. I want to learn how to make these conversations easier for people."<br>"Wanting to feel more comfortable having conversations with dying patients and their family and loved ones." | 36 | 4.6 |
| Discussion/Network | To have an opportunity to discuss death and dying in a safe environment with like-minded people | "I'd like to know some of the ways death and dying are looked at in today's world. The only conversations I have had so far have been with my partner (who died some years ago) and with a friend (whose mother died a few years ago). I would like to be part of a space and discourse where there is no fear attached and where there is only respect, learning and acceptance."<br>"I hope to gain new perspective and ways of thinking about death and dying. Being an online course, (there) will be so many different cultures and global views of death that I may not had the opportunity to connect with otherwise." | 32 | 4.1 |

**Table A2.** *Cont.*

| Theme | Definition | Examples | Number of Participants with This Theme Present in Their Response | Percentage of Participants with This Theme Present in Their Response (%) |
|---|---|---|---|---|
| Cultural | To learn more about how people from different cultures view death and dying, and the different ways of dealing with death | "I hope to discuss more about death and dying and maybe see how other cultures deal with these issues. I hope to understand more the process of grief and mourning- and for that I need to discuss more about death and dying." "How others feel and handle death, how he communities we live in handle death. How best we can help each other handle death better. As a multi-cultural country if we can also learn about other cultures death rituals and their understandings about death and how we can fuse all this together and cater for our aged in our work places be it aged care, hospice or hospital." | 29 | 3.7 |
| MOOC/Online | To experience online learning or to participate in a MOOC specifically | "Know what a MOOC is! Get ideas and inspiration for my work and my life." "More confidence in undertaking online learning and more knowledge about death and dying" "Ways people approach death and dying in a dignified and graceful way. Tips to help me prepare for aging and death. Appreciate the linking of education, technology, staff and students in the MOOC way." | 28 | 3.6 |
| Confidence | To feel confident with talking about dying and to engage people who are at the end of their lives. The main aim of the participant is to eliminate feelings of anxiety when conducting such conversations and thus be better conversationalists, with the end goal of being more competent at it from a work perspective. | "Increased confidence when discussing death and palliative care options with patients and colleagues in the workplace and potentially family members as well." "I hope to come away with more confidence discussing dying with residents and their loved ones." "I hope to be become competent and confident enough to talk about death and the dying process" | 21 | 2.7 |
| Resources | To gain access to sources of information or tools associated with dealing with death | "Learn of new resources, meet and connect with others (hopefully some local people to), to have my thinking challenged." "Tools to improve the death experience for families and their loved ones." "More resources, networking with like-minded and the opportunity to learn from the wisdom being shared." | 13 | 1.7 |
| NIL | (not sure, as above, blank, no expectations) | N/A | 40 | 5.1 |

**Table A2.** *Cont.*

| Theme | Definition | Examples | Number of Participants with This Theme Present in Their Response | Percentage of Participants with This Theme Present in Their Response (%) |
|---|---|---|---|---|
| Free | The course is free of charge for participants | "Because I am interested and because this is a free course, my training budget for 2017 is completely accounted for. This is also a format that is manageable in my busy life."<br>"I have been looking for further education in this area and I thought this sounded interesting and like something I would enjoy, it was right up my ally, It will help get my education points up, and it was FREE" | 27 | 3.4 |
| CPD (Continuing Professional Development) | Motivation was specific to gaining CPD points from participating in the course | "I was looking for something that would add to my CPD points that was relevant to my work but more importantly enhance my insight and performance into the what I feel is the most important part of my work, supporting individuals and families through the end of life process."<br>"Looked interesting, well set up, interactive and adds to my learning and CPD points" | 15 | 1.9 |
| Recommended | Participation in the MOOC was due to a recommendation from a contact | "It was recommended by a colleague who I respect and she sent link-interested in the subject matter feel passionate about I'm proving communication and people and families and dying especially in public hospital setting. Also interested as an educator and how online learning and this subject can work effectively together. Ready to learn new ideas, and new ways of teaching"<br>"This course has come recommended to me by my facilities educator. I like that it is online and can be completed in my own time at my own pace. I also love that the group has many people with different jobs from nurses to carers giving various aspects on death and dying." | 15 | 1.9 |
| Advance Care Planning/Palliative | Motivated by an interest in learning more about ACP/Palliative Care specifically | "I like to keep informed on any new information in Palliative Care to help me support my patients, family or friends more affectively"<br>"I am interested in learning more about ACP (Advanced Care Planning) and how to go about discussing end of life with patients. To give me more ways to discuss and learn about dying." | 14 | 1.8 |
| Comfortable | Seeking to feel more comfortable in conducting conversations related to death and dying | "To help to become more confident with open discussions about death and dying"<br>"I want to feel more comfortable when talking to others about death and dying. I work in residential care and people often say they have moved here to die or living in 'God's waiting room'" | 9 | 1.1 |
| NIL | (no response given) | N/A | 11 | 1.4 |

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
