# Peer review of "Participation in an Online Course about Death and Dying: Exploring Enrolment Motivations and Learning Goals of Health Care Workers"

_education, doi:10.3390/educsci10040112_

Round 1

Reviewer 1 Report

Thank you for a interesting paper!

I miss a more in depth description of the course content and curriculum in the introduction/methods section. 

Is communication part of the curriculum? This should be discussed more in depth as this was a common learn ing goal for HCWs.

The small n umber of doctors should be included in the limitations. 

Reviewer 2 Report

This research investigates the reasons and the goals of health-care workers for their enrollment in a MOOC on death and dying. Qualitative data on their views were collected and themes were identified. Results indicate that most of them were motivated by a general interest in death and dying, and by work-related reasons. There are many interesting findings such as gaining confidence is more common among younger health care workers as their goal in the enrollment of the course, and the mode of course delivery (i.e. online) is a reason for their enrollment (line 238 – 239). These findings undoubtedly contribute to existing knowledge on death and dying.

The research methods are clearly described in details and inter-rater reliability were adopted to enhance the reliability of the coding process. However, collecting quantitative data instead of qualitative data only can strengthen the results of this study.

There is considerable overlapping in the themes identified for the two research questions. Reporting the findings in two separate long tables that list the themes with examples might not be very helpful to the readers in understanding the results. One suggestion is to put the two tables (Table 3 and 4) as appendices and highlight the results of these two tables in a short and more informative table.

A limitation of this research is that the participants were mostly female health care workers. This profile of female constituting 93.3% of the sample may render the gender differences identified from the statistics (e.g. line 125 – 129; line 182 – 186; line 249 – 250) not very informative. There is also a tendency for the authors to make interpretations that are not strong in addition to the above. Some examples are given below:

  1. As most participants (about 75%) were from the health-related professions, it can be expected that work-related motivations and goals were the most popular among participants. The interpretation given in line 235 – 236 may seems to be too simplistic to the readers. Similarly, readers may find the finding of non-health-care workers being more motivated by personal reasons (line 198 – 199; line 216 – 217) simplistic.
  2. A conclusion made by the author is that the work-related motivation of health care workers for enrolling in the course reflects that inadequate training on this area. However, it can be argued that health care workers might be motivated by the importance of death and dying in their work instead of insufficient training.
  3. Since few doctors attended the course, readers may not be very interested in the results reported between line 318 and 331. The authors can consider removing section 4.3.

Most readers would not expect results especially in the form of verbatim to be included in the discussion section instead of the result section. The authors can consider removing the verbatims in section 4.2 which make the discussions too long and difficult to follow.

It is mentioned in the last sentence (line 345 – 346) that “This knowledge could inform future development of similar courses or resources targeted at HCWs professional training”. I would suggest more specific directions to be included.

Reviewer 3 Report

Well designed and developed study about an open online course on death and dying. In terms of conclusions, I think that the authors can improve, if they present some clues for the future, what contributions they can make with this research for the field of death education, health and education in general.

Author Response

We've amended the conclusion to include some specific suggestions to how the study findings could inform future development of courses targeted at health care workers

Reviewer 4 Report

The article addresses an original topic of great interest for the training of health professionals, through an open and accessible educational proposal as a MOOC. The study is rational and rigorous, regarding its objectives and methodology.

However, it needs for further study, especially in the theoretical review, as well as in the implications of the results obtained:

- The theoretical revision of the method is poore. An analysis of concepts, models and research on at least two fundamental constructs is required: 'death education', and the MOOC as an online platform for learning.
- This broader review could help to include in the discussion other studies or research with which to compare the results obtained.
- Further development is needed abouth the implications of the results for the design of training actions on death education with health professionals.

Some studies on death education that could be reviewed are:

Affifi, R., and B. Christie. 2018. “Facing Loss: Pedagogy of Death.” Environmental Education Research. Published online: 02 Mar 2018.

Bibeau, D. & Eddy, J. M. (1985). The effect of death education course on dying and death knowledge, attitudes and fears. Health Educator, 17 (1), 15-18.

Birkholz, G., P. Clements, R. Cox, and A. Gaume. 2004. “Students’ Self-identified Learning Needs: A Case Study of Baccalaureate Students Designing their Own Death and Dying Course Curriculum.” Journal of Nursing Education 43 (1): 36-39.

Corr, C., D. Corr, and K. Doka. 2019. Death and Dying, Life and Living. Pacific Grove, CA: Brooks/Cole Publishing Company.

Henoch, I., Melin-Johansson, C., Bergh, I., Strang, S., Ek, C., Hammarlund, K., Lundh Hagelin, C., Westin, L., Österlind, J. & Browall, M. (2017). Undergraduate nursing students' attitudes and preparedness toward caring for dying persons - A longitudinal study. Nurse Education in Practice, 26, 12-20.

Lee, H., Jo, K., Chee, K., & Lee, Y. (2008). The perception of good death among human service students in South Korea: a Q-Methodological approach. Death Studies, 32, 870-890.

Kim, Y., Ahna, S. Y., Leea, C. H., Leeb M. S., Kimc, M. J., Armac, P., Hwangd, H. J., Songe, H. D., Shimf, M. S. & Kima, K. H. (2016). Development of a death education curriculum model for the general public using DACUM method. Technology and Health Care, 24, 439-446.

Zhao, S., Qiang, W., Zheng, X. & Luo, Z. (2018). Development of death education training content for adult cancer patients: A mixed methods study. Journal of Clinical Nursing, 27, 4400-4410.

Round 2

Reviewer 2 Report

My responses to authors' revisions based on my suggestions in my previous reviews are given in red in the attached file.

Reviewer 4 Report

Thank you for considering the suggestions.